

# Confero
## Wsparcie użytkowników hybrydowych konferencji naukowych.

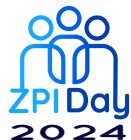

**Autorzy**: Dzianis Klebanovich · Artsiom Shablinski · Uladzislau Zherabiatsyeu
· Vera Yemialayanva

**Opiekun:** Zbigniew Telec

### Streszczenie

Projekt „Wsparcie użytkowników hybrydowych konferencji naukowych" ma na celu uproszczenie organizacji wydarzeń naukowych najwyższej rangi (A i A* wg CORE) poprzez zrobienie zaawansowanej platformy cyfrowej. Inicjatywa rozwiązuje wyzwania związane z zarządzaniem sesjami specjalnymi, zapewniając kompleksowe wsparcie zarówno dla osób zgłaszających propozycje sesji, jak i administratorów odpowiedzialnych za ich weryfikację i akceptację.

Główne cele projektu to automatyzacja procesów organizacyjnych, w tym usprawnienie zgłaszania sesji specjalnych dla ich autorów oraz zapewnienie administratorom czytelnego systemu weryfikacji propozycji, umożliwienie uczestnikom interakcji w czasie rzeczywistym poprzez system wideokonferencji oraz czatu, wprowadzenie możliwości łatwego zakładania profili użytkowników z wykorzystaniem zewnętrznych dostawców tożsamości, zapewnienie osobom występującym narzędzi do przesyłania i udostępniania materiałów związanych z ich wystąpieniami oraz zarządzanie harmonogramem sesji. Dzięki temu rozwiązaniu organizacja hybrydowych konferencji stanie się prostsza i bardziej efektywna.

## 1   WSTĘP

W organizacji hybrydowych konferencji naukowych istniało kilka kluczowych wyzwań, które wymagały rozwiązania. Pierwszym z nich był proces zgłaszania sesji specjalnych przez uczestników. W tradycyjnym modelu, uczestnicy musieli ręcznie wypełniać i edytować dokumenty Word zgodnie z określonym schematem, a następnie wysyłać je do administratorów w celu weryfikacji. Proces ten wiązał się z częstą wymianą mailową i dużym ryzykiem popełniania błędów. Kolejnym problemem było ręczne wpisywanie danych uczestników sesji, co również niosło za sobą ryzyko pomyłek. Konieczne stało się znalezienie rozwiązania, które umożliwiłoby integrację z zewnętrznymi bazami danych o naukowcach, takimi jak ORCID, aby zautomatyzować proces weryfikacji.Dodatkowym wyzwaniem była potrzeba efektywnego zarządzania słowami kluczowymi przypisanymi do sesji specjalnych, co wymagało usprawnienia poprzez generowanie słów kluczowych na podstawie istniejących słowników.

Istotnym problemem była również komunikacja pomiędzy uczestnikami oraz prelegentami, szczególnie w kontekście hybrydowych konferencji, gdzie część uczestników brała udział w wydarzeniu stacjonarnie, a inni zdalnie. Dodatkowo, dla osób prezentujących, należało umożliwić umieszczanie własnych materiałów do wystąpień w postaci różnych plików. Ważnym elementem było także opracowanie dedykowanej strony dla każdej sesji specjalnej, zawierającej szczegółowe informacje na temat sesji, umożliwiając uczestnikom zapoznanie się z harmonogramem i najistotniejszymi danymi przed wydarzeniem.

**Główne cele projektu obejmowały:**

- Automatyzację procesu zgłaszania sesji specjalnych poprzez formularz online oraz śledzenie ich statusu, eliminując wymianę mailową.

- Integrację z ORCID API, co pozwoliło na automatyczne pobieranie danych uczestników, eliminując błędy wynikające z ręcznego wpisywania danych.

- Umożliwienie komunikacji między uczestnikami w czasie rzeczywistym, poprzez czat i wideokonferencje.

- Integrację z zewnętrznymi providerami tożsamości (Google, LinkedIn, ORCID), aby umożliwić tworzenie profili użytkowników bez konieczności ich ręcznego zakładania.

## 2 PRACE POWIĄZANE

### 2.1 Przegląd istniejących rozwiązań

W ramach analizy dostępnych rozwiązań porównano projekt z platformą Whova, szeroko stosowaną w organizacji wydarzeń hybrydowych. Whova oferuje funkcje zarządzania agendą, networking uczestników i interaktywne sesje, jednak nie wspiera procesów zgłaszania i weryfikacji sesji specjalnych ani integracji z bazami akademickimi, jak ORCID. Projekt „Wsparcie użytkowników hybrydowych konferencji naukowych" wyróżnia się koncentracją na potrzebach środowiska akademickiego. Automatyzuje procesy weryfikacji, zarządzania słowami kluczowymi i integracji z systemami tożsamości. W porównaniu do Whova, dostarcza narzędzie lepiej dostosowane do specyfiki konferencji naukowych, wypełniając istniejące luki w dostępnych platformach.

### 2.2 Założenia projektowe

Projekt został zrealizowany w ramach ograniczonego okresu dwóch miesięcy jako wersja MVP (Minimum Viable Product), koncentrując się na implementacji kluczowych funkcjonalności, które umożliwiły weryfikację wartości i użyteczności rozwiązania.

### 2.3 Technologie

W projekcie wykorzystano nowoczesne technologie, takie jak Spring Boot na backendzie, React na frontendzie oraz PostgreSQL jako system zarządzania bazą danych. W celu budowy aplikacji zastosowano narzędzia Gradle i npm, natomiast do wdrożenia produkcyjnego użyto Docker oraz reverse proxy NGINX. Biblioteka komponentów shadcn, oparta na Tailwind CSS, przyspieszyła proces tworzenia interfejsów użytkownika. Projekt opierał się na integracjach z zaawansowanymi narzędziami, takimi jak Supabase (autoryzacja, S3, hosting baz danych), TextRazor (generowanie tagów z opisów sesji), ORCID API (automatyzacja weryfikacji danych naukowców) oraz @jitsi/react-sdk (obsługa strumieniowania wideo). Przyjęto podejścia API-first oraz Database-first, co pozwoliło na przygotowanie specyfikacji OpenAPI 3 i automatyczne generowanie elementów kodu, takich jak kontrolery backendowe oraz klient API frontendowy. Podczas etapu deweloperskiego duży nacisk położono na jakość kodu, pisząc liczne testy integracyjne dla backendu z wykorzystaniem narzędzi REST Assured, JUnit, Mockito oraz Testcontainers. Dzięki temu zespół mógł wcześnie identyfikować i eliminować potencjalne błędy. Procesy CI/CD zostały w pełni zautomatyzowane za pomocą GitHub Actions, co zapewniło sprawny przepływ pracy i efektywne wdrożenia.

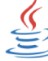 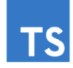 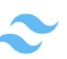 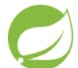 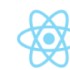 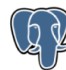 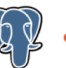 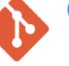 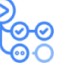 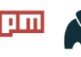 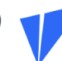 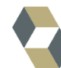 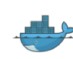 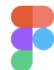

## 3 REZULTATY

### 3.1 Zrealizowane funkcjonalności

**System autoryzacji**
Standardowa autoryzacja za pomocą adresu e-mail i hasła oraz autoryzacja za pomocą *Google*, *LinkedIn*, *ORCID* pozwala na łatwy i bezpieczny dostęp do naszej aplikacji.

**Profil użytkownika**
Profil użytkownika, który pozwala na zarządzanie swoimi adresami e-mail oraz ORCID. Istnieje możliwość dodania kilku adresów e-mail oraz zatwierdzanie ORCID.

**Zgłoszenie propozycji**
System pozwala na zgłoszenie propozycji sesji za pomocą UI oraz JSON. Przy zgłoszeniu za pomocą UI istnieje możliwość wygenerowania słów kluczowych na podstawie opisu zgłoszenia. Dane osób prezentujących pobierane są za pomocą wpisanego ORCID, co gwarantuje poprawność danych. Istnieje możliwość zapisania zgłoszenia jako szkic, aby uzupełnić pozostałe informacje później.

**Ocenianie zgłoszeń**

Po złożeniu propozycji sesji, administrator widzi zgłoszenie w systemie i ma możliwość zaakceptowania, odrzucenia albo pozostawienia komentarza z prośbą o dostosowanie zgłoszenia. Gdy administrator poprosi o dostosowanie, osoba zgłaszająca ma możliwość poprawienia zgłoszenia i ponownego przesłania do oceny.

**Wyświetlanie sesji**

Po zatwierdzeniu zgłoszenia, sesja staje się widoczna na stronie głównej. Osoby niezalogowane i zalogowane mają możliwość oglądania podstawowych informacji o sesji i harmonogramu. Osoby zalogowane mają możliwość uczestnictwa w sesji, czyli dołączenia do wideokonferencji, zadawania pytań na czacie oraz pobrania plików dotyczących prezentacji.

**Kalendarz**

System pozwala na dodanie sesji do kalendarza oraz wyeksportowania w formacie .ics, co pozwala dodać rozkład sesji do kalendarza na swoim urządzeniu.

**Zarządzanie prezentacjami**

Osoba prezentująca ma możliwość dodawania plików, ustawienia harmonogramu oraz edycji informacji o prezentacji.

**Wyszukiwanie organizatorów**

Istnieje możliwość wyszukiwania organizatorów oraz dodania sesji organizatorów do swojego kalendarza.

**Zarządzanie edycjami konferencji**

Administrator ma możliwość edycji oraz tworzenia edycji konferencji, ustawienia terminu ostatecznego zgłoszenia aplikacji oraz wgrania listy zaproszonych w postaci pliku CSV.

## 3.2 Osiągnięte cele biznesowe oraz techniczne

**Uproszczenie procesu organizacji konferencji naukowych**

- Stworzona platforma zautomatyzowała kluczowe procesy organizacyjne, takie jak zgłaszanie i weryfikacja sesji specjalnych, eliminując konieczność ręcznego zarządzania dokumentami.

- Zmniejszono czas potrzebny na obsługę zgłoszeń i komunikację między organizatorami a uczestnikami, co pozwala na lepsze wykorzystanie zasobów ludzkich.

**Zwiększenie satysfakcji użytkowników**

- Uczestnicy uzyskali możliwość łatwego zgłaszania sesji poprzez intuicyjny interfejs użytkownika oraz możliwość korzystania z zewnętrznych dostawców tożsamości (*Google*, *LinkedIn*, *ORCID*), co przyspieszyło rejestrację i ograniczyło błędy.

- System umożliwił uczestnikom interakcję w czasie rzeczywistym za pomocą wideokonferencji i czatu, co poprawiło jakość doświadczenia użytkowników w hybrydowym środowisku.

**Lepsza promocja konferencji i zaangażowanie uczestników**

- Dedykowane strony dla każdej sesji specjalnej oraz możliwość eksportu harmonogramu do kalendarza ułatwiły uczestnikom planowanie i zaangażowanie w wydarzenia.

- Zautomatyzowane generowanie słów kluczowych ułatwiło zgłoszenie propozycji, eliminując potrzebę ręcznego dodawania.

## 4 WNIOSKI

## 4.1 Osiągnięte wyniki

Projekt został pomyślnie zrealizowany, osiągając założone cele i dostarczając funkcjonalności kluczowe dla uczestników i organizatorów konferencji, a mianowicie:

**Zwiększenie dostępności konferencji hybrydowych**

Aplikacja umożliwia zarówno uczestnikom stacjonarnym, jak i zdalnym, pełny dostęp do materiałów i harmonogramu konferencji, co znacząco zwiększa wygodę i zasięg wydarzenia.

**Sprawna organizacja i zarządzanie konferencją**

Wdrożenie funkcjonalności do rejestracji zgłoszeń oraz ich zatwierdzania przez administratorów uprościło proces zarządzania treścią konferencji. Funkcjonalność edycji harmonogramu sesji pozwala na szybkie dostosowanie planu do dynamicznie zmieniających się potrzeb.

**Wsparcie dla występujących i uczestników**

Możliwość przesyłania plików przez występujących usprawniła przygotowania do sesji i zapewniła łatwy dostęp do materiałów dla uczestników. Uczestnicy otrzymali intuicyjny dostęp do wideo i dokumentów, co zwiększyło ich zaangażowanie i satysfakcję.

**Łatwość dostępu dla użytkowników anonimowych**

Publiczny dostęp do harmonogramu i szczegółów sesji bez konieczności logowania zwiększył transparentność i zachęcił większą liczbę osób do udziału w konferencji.

**Logowanie bez barier: szybki dostęp do systemu**

Wdrożono funkcję logowania i rejestracji użytkowników przy użyciu popularnych aplikacji zewnętrznych, takich jak ORCID, Google, LinkedIn, czy poczta e-mail. Dzięki tej funkcjonalności użytkownicy mogą korzystać z aplikacji w sposób szybki i bezpieczny, bez potrzeby tworzenia dodatkowych kont. To rozwiązanie usprawnia proces weryfikacji danych (np. danych naukowych przez ORCID) i zwiększa wygodę korzystania z systemu.

**Spójna obsługa wieloletnich konferencji**

Aplikacja została zaprojektowana w sposób umożliwiający wykorzystanie jej dla kolejnych edycji tej samej konferencji. To rozwiązanie umożliwia organizatorom zachowanie ciągłości danych, co może być cenne zarówno dla statystyk, jak i promocji kolejnych konferencji.

## 4.2 Najważniejszy sukces projektu

Najważniejszym osiągnięciem projektu jest zaprojektowanie kompleksowego rozwiązania, które w prosty sposób łączy potrzeby różnych grup użytkowników (organizatorów, występujących, uczestników) w jednym systemie. Dzięki temu Oorganizatorzy mogą efektywnie zarządzać konferencją, co oszczędza czas i minimalizuje ryzyko błędów. Uczestnicy hybrydowych wydarzeń uzyskali komfortowy i zintegrowany sposób dostępu do treści konferencji, bez względu na miejsce ich przebywania. Występujący zyskali intuicyjne narzędzia do zarządzania swoimi wystąpieniami, w tym możliwość dodawania materiałów takich jak prezentacje i dokumenty. Dzięki temu mogą lepiej przygotować się do swoich sesji, a ich treści są łatwo dostępne dla uczestników. Funkcjonalność ta pozwala na podniesienie jakości wystąpień i ułatwia dzielenie się wiedzą w sposób profesjonalny i uporządkowany.

## 4.3 Znaczenie dla odbiorcy biznesowego i technologicznego

**Biznesowo:** Aplikacja zwiększa atrakcyjność i konkurencyjność organizowanych konferencji, co może przełożyć się na większe zainteresowanie i liczbę uczestników.

**Technologicznie:** Rozwiązanie opiera się na różnorodnych zaawansowanych mechanizmach np. integracja z systemami zewnętrznymi i zarządzanie danymi w czasie rzeczywistym, co gwarantuje skalowalność oraz elastyczność rozwoju.

# 5 PRZYSZŁE KIERUNKI ROZWOJU

**Rozszerzenie wsparcia dla występujących:**

- Funkcja automatycznego tworzenia transkrypcji wystąpień.

- Integracja z dodatkowymi systemami do streamingu i nagrywania prezentacji.

**Rozbudowanie funkcji analitycznych:**

- Statystyki dotyczące liczby uczestników na poszczególnych sesjach.

- Analizy zaangażowania uczestników, np. na podstawie pytań i interakcji.

**Rozszerzenie personalizacji doświadczeń użytkownika:**

- Rekomendacje sesji na podstawie zainteresowań uczestnika.

- Powiadomienia przypominające o nadchodzących wydarzeniach.

**Obsługa wielojęzyczności:**

- Tłumaczenie interfejsu i materiałów konferencyjnych na różne języki.

- Automatyczne tłumaczenie napisów podczas sesji.

**Wsparcie dla większych wydarzeń:**

- Skalowanie aplikacji do obsługi równoległych konferencji.

- Funkcjonalność łączenia prezentacji z różnych konferencji w ramach jednej sesji.

- Rozwój platformy w kierunku dużych kongresów międzynarodowych.

**Ulepszenie zarządzania danymi:**

- Wprowadzenie zaawansowanego systemu archiwizacji danych poprzednich edycji konferencji.

- Lepsza integracja z platformami CRM i narzędziami do marketingu.

**Wdrożenie systemu monetyzacji:**

- Możliwość sprzedawania biletów i pakietów uczestnictwa bezpośrednio przez platformę.

- Model subskrypcyjny dla organizatorów, oferujący dodatkowe funkcjonalności premium.

**Integrację z platformami e-learningowymi:**

- Tworzenie trwałych baz wiedzy dla uczestników i występujących.

# 6   PODZIĘKOWANIA

Chcielibyśmy podziękować wszystkim osobom zaangażowanym w projekt „Wsparcie uczestników hybrydowych konferencji naukowych". dr inż. Krystian Wojtkiewicz – Dziękujemy za analizę rynku, określenie profilu użytkownika i priorytetyzację potrzeb klientów. Pomogło nam to lepiej dostosować funkcjonalności do realnych potrzeb.

dr inż. Zbigniew Telec – Dziękujemy za motywację i pomoc w nawiązywaniu kontaktów z ekspertami, co znacząco przyczyniło się do sukcesu projektu. dr inż. Rafał Palak – Dziękujemy za wnikliwą analizę projektu i identyfikację błędów. dr hab. inż. Marek Krótkiewicz, prof. uczelni – Dziękujemy za wsparcie w projektowaniu aplikacji i zrozumieniu potrzeb klientów. dr inż. Marcin Jodłowiec – Dziękujemy za wskazanie błędów i zaangażowanie w doskonalenie projektu. Pańska analiza pomogła podnieść jakość aplikacji.

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
