# OpenReview forum: "Confero"
_pwr.edu.pl/Wrocław_University_of_Science_and_Technology/2024/ZPI_Day — Wrocław University of Science and Technology 2024 ZPI Day Submission_

### Official Review · Reviewer_4wAv · 2024-12-04
**Wartościowa praca wypełniająca funkcjonalności organizacji wydarzeń naukowych najwyższej rangi - CORE**

**Confidence:** 5
**Significance Of Results:** 4
**Overall Quality:** 4

**Compliance With Template:**

5: Very High Quality – The article contains all the required sections, which are written in a very detailed, clear, and error-free manner. The structure is professional and meets expectations, and the content adheres to the highest substantive and formal standards.

**Description Of Results:**

4: High Quality – The results are described in detail and supported by usage examples or evaluations. The description is reliable but may lack full depth of analysis.

**Feedback On Consistency:**

Artykuł napisany popranie i spójnie językowo. Cel pracy jasno zdefiniowany oraz zrealizowany.

**Potential For Development:**

Praca posiada duży potencjał do dalszego rozwinięcia oraz jego komercjalizacji.

**Project Nature Evaluation:**

Praca spełnia w całości wymogi stawiane pracom inżynierskim. Opracowanie rozwiązuje ważny problem z organizacją wydarzeń naukowych najwyższej rangi CORE A* oraz A. Poprzez wykonanie zaawansowanej platformy webowej wspomagającej przepływ dokumentów oraz organizację sesji naukowych.
Technologie dobrane prawidłowo do opracowywanej tematyki. Nowatorskie podejście do tematu widoczne w implementacji czatu i widokonferencji w samej aplikacji.
Pewnym brakiem w artykule jest całkowity brak zdjęć przedstawiających opracowany system. Opisy rezultatów cechuje średnia szczegółowość.

**Technical Language Precision:**

4: High Quality – The language is appropriate for a technical report. Terminology is used correctly, and statements are precise, with only minor shortcomings that do not affect the overall clarity.

---

### Official Review · Reviewer_m4Xy · 2024-12-05
**Confero - Wsparcie użytkowników hybrydowych konferencji naukowych.**

**Confidence:** 5
**Significance Of Results:** 5
**Overall Quality:** 4

**Compliance With Template:**

5: Very High Quality – The article contains all the required sections, which are written in a very detailed, clear, and error-free manner. The structure is professional and meets expectations, and the content adheres to the highest substantive and formal standards.

**Description Of Results:**

4: High Quality – The results are described in detail and supported by usage examples or evaluations. The description is reliable but may lack full depth of analysis.

**Feedback On Consistency:**

The project goals are clearly stated what influenced the consistemncy of the approach to the project. As the result the description of the project is also consisten. The motivation for the product creation, the analysis of the subjected problem as well as presentation of the results and conlcusions are presented very well. However, there is no information in the paper about the approach to verifying the quality of the product.

**Potential For Development:**

The paper containd rich information about plans for product evolution. They are stated explicitly in the appropriate part of the document. Also the practical aspects of the project are clearly described – the product may be applied for one conference first, and then for the sequence of the same conference, for many conferences and even for interaction with several confeneces at the same time.

**Project Nature Evaluation:**

The project is on a high technical level – it is very useful, new technological solutions were applied in the product and some functionalities are supported by external services. The technologies applied are described very well. It seems from the description that good technical decision were made by the project team.

**Technical Language Precision:**

5: Very High Quality – The language is entirely appropriate for a technical report. All terms are used correctly and precisely, and the style is professional, clear, and coherent, without any errors or ambiguities.

---

### Official Review · Reviewer_P59u · 2024-12-05
**Wsparcie użytkowników hybrydowych konferencji naukowych**

**Confidence:** 5
**Significance Of Results:** 2
**Overall Quality:** 2

**Compliance With Template:**

2: Low Quality – The article includes only some of the required sections, and their content is superficial, underdeveloped, or difficult to understand. Some sections are entirely omitted or insufficiently elaborated.

**Description Of Results:**

2: Low Quality – The results are described very superficially and in a general manner. Essential details, usage examples, or evaluations are missing.

**Feedback On Consistency:**

Projekt został przedstawiony bardzo powierzchownie. Prezentacja wyników sprowadza się do wypunktowania funkcjonalności.

**Potential For Development:**

Projekt ma potencjał wdrożeniowy.

**Project Nature Evaluation:**

Projekt wykazuje cechy pracy inżynierskiej, ale prezentacja wyników została, źle wykonana.

**Technical Language Precision:**

3: Average Quality – The language is mostly appropriate but may contain minor terminological or stylistic errors. Some statements might lack precision or require improvement for better readability.

---

### Decision · Program_Chairs · 2024-12-10

Accept (Poster)